



# Future changes in North Atlantic winter cyclones in CESM-LE - Part 2: A Lagrangian analysis

Edgar Dolores-Tesillos[1,2] and Stephan Pfahl[1]

[1]Institute of Meteorology, Freie Universität Berlin, Berlin, Germany
[2]Institute of Geography, Oeschger Centre for Climate Change Research, University of Bern, Bern, Switzerland

**Correspondence:** Edgar Dolores-Tesillos (edgar.dolores@unibe.ch)

**Abstract.**

Future changes in extratropical cyclone structure and dynamics may lead to important impacts, but are not yet fully understood. In the first part of this study, we have applied a composite approach together with potential vorticity (PV) inversion to study such changes in the dynamics of North Atlantic cyclones. Here, this is complemented with the help of a Lagrangian perspective, making use of air parcel trajectories to investigate the causes of altered PV anomalies as well as the role that cyclone airstreams play in shaping these changes. Intense cyclones in the extended winter seasons of two periods, 1990-2000 and 2091-2100, are studied in Community Earth System Model Large Ensemble (CESM-LE) simulations, and backward trajectories are calculated from the cyclone area as a basis to construct cyclone-centered composites of Lagrangian tendencies and their projected future changes. Our results show that diabatic processes on a timescale of 24 hours shape the cyclones' low-level PV distribution and corroborate that the increasing moisture content along with enhanced ascent in warm conveyor belts leads to amplified latent heat release and larger low- and mid-level PV anomalies near the cyclone center in a warmer climate. In contrast, projected upper-level PV changes are due to a combination of several processes, in addition to cloud-diabatic PV changes mainly anomalous PV advection and likely also radiative PV generation in the lower stratosphere above the cyclone center. For instance, enhanced poleward advection is the primary reason for a projected decrease in upper-level PV anomalies south of the cyclone center. Warm conveyor belt outflow regions are projected to shift upward, but there is not robust change in the associated upper-level PV anomalies due to compensation between enhanced low-level PV generation and upper-level PV destruction. In summary, our 2-part study points to future changes in the relative importance of different processes for the dynamics of intense North Atlantic cyclones in a warming climate, with important consequences for the near-surface wind pattern. In particular, a larger role of cloud diabatic processes is projected affecting the cyclones through PV production in the lower troposphere. The role of other mechanisms, in particular radiative changes near the tropopause, should be investigated in more detail in future studies.

## 1 Introduction

European windstorms are typically associated with extratropical cyclones traveling over the North Atlantic towards Europe, with higher cyclone frequencies and intensities in the winter compared to the summer half-year (Leckebusch et al., 2006;

 

Laurila et al., 2021; Ulbrich et al., 2009). The strong winds associated with these storms can result in large societal impacts and economic losses (Klawa and Ulbrich, 2003; Roberts et al., 2014). For instance, the death of 65 people across Europe has been attributed to the extratropical cyclone Xynthia (Kolen et al., 2010). Cyclones, through damaging winds, also cause 53% of all damages to European forests (Schelhaas et al., 2003). To be able to adapt to potential future changes in European windstorm intensities, it is thus crucial to understand and predict changes in extratropical cyclone dynamics (cf. Shaw et al., 2016; Catto
et al., 2019).

In part 1 of this study (Dolores-Tesillos et al., 2022, , in the following simply referred to as part 1), we have analyzed such changes in cyclone dynamics based on Community Earth System Model Large Ensemble (CESM-LE) simulations for the RCP8.5 scenario and found an intensification of near-surface wind speed and an extended wind footprint in the warm sector southeast of the cyclone center during the mature cyclone stage. Such an extended footprint has also been found for
several CMIP6 models and other climate scenarios (Priestley and Catto, 2022). Furthermore, based on the analysis of potential vorticity (PV) anomalies and PV inversion, we have demonstrated that both enhanced diabatic heating and changes in upper-level dynamics (represented by lower- and upper-level PV anomalies, respectively) contribute to the near-surface wind changes. In part 2, we go one step further by investigating the processes leading to changes in these PV anomalies through a Lagragian analysis of cyclone airstreams.

Several studies have used the concept of cyclone-relative airstreams to describe the cloud organization in extratropical cyclones (Harrold, 1973; Carlson, 1980; Browning, 1990). By analyzing the system-relative flow on isentropic surfaces around mid-latitude cyclones, the presence of three distinct airstreams has been demonstrated: the warm conveyor belt (WCB), the dry intrusion (DI), and the cold conveyor belt (CCB) (Harrold, 1973; Carlson, 1980; Browning, 1986; Madonna et al., 2014). Fig. 1 shows the region and typical trajectory of the three airstreams. A detailed perspective on these airstreams and their
climatological characteristics can be obtained from air parcel trajectory calculations (e.g., Wernli and Davies, 1997; Wernli, 1997; Madonna et al., 2014; Raveh-Rubin, 2017).

WCBs are moist airstreams originating from near the surface ahead of cold fronts and experiencing rapid slantwise ascent with latent heat release. In this way, they provide a major contribution to cloud and precipitation formation in extratropical cyclones (Madonna et al., 2014). Furthermore, they play a crucial role in extreme precipitation events in the mid-to-high
latitudes (Pfahl et al., 2014). Embedded convection in the WCB may contribute to such high precipitation rates (Oertel et al., 2020). Joos (2019) demonstrate the relevance of WCBs for modulating the radiative budget in mid-latitudes. For instance, they contribute around 10 $Wm^2$ to the net cloud radiative forcing over the central North Atlantic in winter. In the Northern Hemisphere, WCBs are more frequent in winter than in summer, with two preferential regions of ascent in the western North Atlantic and western North Pacific (Madonna et al., 2014), typically fueled by moisture from nearby ocean evaporation (Pfahl
et al., 2014).

WCBs influence the tropospheric PV distribution in different ways. Before the ascent, the airstreams, on average, have relatively low PV, followed by a steep PV increase during the first part of the ascent and a decrease in the second part (Madonna et al., 2014). This PV ascent and descent are due to the latent heating during various microphysical processes, such as condensation of water vapor at low levels and depositional growth of snow at upper levels (Binder et al., 2016; Joos and Wernli,



2012). WCB-related cloud-diabatic processes thus produce positive PV anomalies in the lower and middle troposphere, which can reinforce the cyclone's intensification (Dacre and Gray, 2013; Binder et al., 2016) and also affect other dynamical features, such as the low-level jet ahead of the cold front (Lackmann, 2002). At upper levels, diabatic PV destruction leads to negative PV anomalies in the WCB outflow near the tropopause level, which can interact with the extratropical waveguide and thereby substantially influence the downstream flow (Binder et al., 2016; Grams et al., 2011). Additionally, this process may also influence the formation of atmospheric blocking downstream (Steinfeld and Pfahl, 2019; Steinfeld et al., 2020).

The CCB is characterized by a westward flow relative to the cyclone propagation at low levels parallel to and on the cold side of the warm front (Catto et al., 2010; Schemm and Wernli, 2014). It often consists of two branches. The main branch slopes cyclonically at low levels around the cyclone center (Schultz, 2001) and is often related to strong winds along the bent-back front, forming a low-level jet (Schemm and Wernli, 2014; Slater et al., 2017). A second branch turns anticyclonically and ascends into the cloud head. Schultz (2001) mentioned that this branch represents a transition airstream between the WCB and the cyclonic branch of the CCB. Similar to the WCB, cloud-diabatic processes in the CCB can produce positive PV anomalies in the lower and middle troposphere (Schemm and Wernli, 2014).

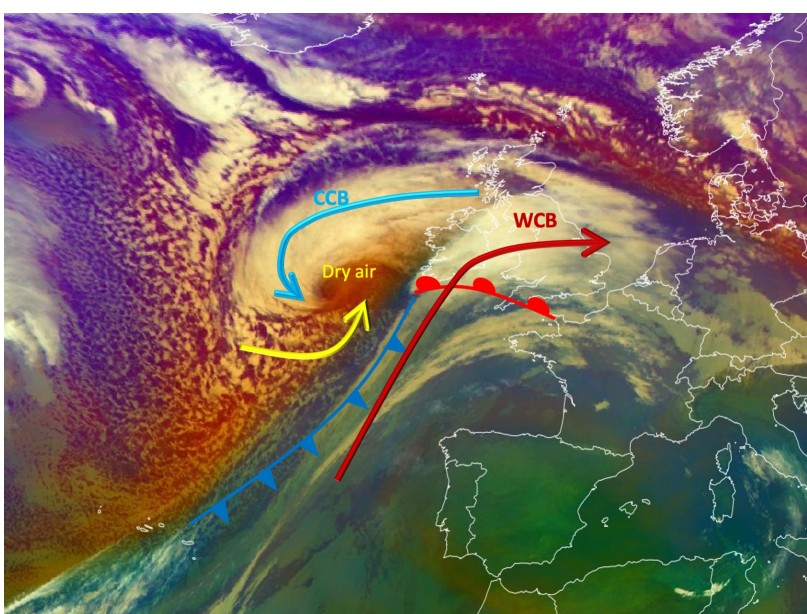

**Figure 1.** Satellite image of the extratropical cyclone Eunice (February 18, 2022) with the three airstreams WCB, DI and CCB indicated by colored arrows. The red and blue lines show the cyclone's warm and cold front, respectively. Modified from: EUMETSAT Eumetview (https://view.eumetsat.int/productviewer). ©EUMETSAT 2022.

DIs are cold, dry airstreams descending behind the cyclone from near the tropopause to middle and low levels (Browning, 1997; Catto et al., 2010; Raveh-Rubin, 2017). At lower levels, DIs tend to fan out cyclonically or anticyclonically. A cloud-free region is generated due to the low moisture content in DIs (Catto et al., 2010; Raveh-Rubin, 2017), which can be identified in satellite images as a "dry slot" with a hammer-head shape (Fig. 1). The cyclonically descending air parcels might travel along





the cold front and produce instability and potential for severe weather (Browning, 1997). Convection associated with the DIs is found in regions where they overrun moist air and potential instability is released, favoring strong winds, precipitation, and thunderstorms (e.g., Raveh-Rubin and Wernli, 2016). However, DIs can also weaken the WCB rain band if they underrun the

northern part of the WCB, leading to evaporation of precipitation from the cloud head (Raveh-Rubin, 2017). DIs influence the PV distribution mainly through advection of air masses with relatively high PV from upper layers downwards (Raveh-Rubin, 2017).

Only very recently, first studies have started to investigate changes of cyclone-related airstreams, in particular, the WCB, in a warmer climate from a Lagrangian perspective. Based on the same CEMS-LE simulations as applied here (see section 2),

Joos et al. (2023) and Binder et al. (2023) studied projected future changes in WCB airstreams as well as their role in cyclone intensification. Joos et al. (2023) concluded that projected shifts in the spatial distribution of WCBs are mostly consistent with shifts in the storm tracks and that the larger atmospheric moisture content in a warmer climate causes a systematic increase in WCB-related precipitation, latent heat release and outflow level. According to Binder et al. (2023), this enhanced latent heating and the associated increase in diabatic PV production lead to an increased relative importance of diabatic processes for rapid

cyclone intensification in a warmer climate. However, at least in the Northern Hemisphere, this is not associated with a general intensification of cyclones compared to present-day climate, most probably due to compensating changes in dry-dynamical processes. These results are consistent with previous findings based on idealized models (Pfahl et al., 2015; Büeler and Pfahl, 2019; Sinclair et al., 2020) and our results from part 1.

In this part 2, we extend the analysis of future changes in cyclone airstreams by combining Lagrangian trajectory calculations

with the Eulerian composite perspective applied in part 1. The main goal is to explicitly identify the processes contributing to the composite PV anomaly changes identified in part 1 and link them to the properties of airstreams, in particular the WCB and DI. In this way, we test the hypothesis (also formulated in part 1) that cloud-diabatic processes are mainly responsible for low-level PV changes, but additional factors have to be considered for explaining the upper-level changes. In sections 2 and 3 below, we describe the CESM-LE dataset as well as the trajectory calculations and generation of Lagrangian composites,

respectively. The results are presented and discussed in section 4, and conclusions are provided in section 5.

## 2  Data

We have selected ten members from the CESM-LE-ETH ensemble, which were restarted from CESM-LE simulations (Kay et al., 2015) proving 6 hourly output fields on model levels that are required for our trajectory calculations (see section 3). The periods 1990-2000 (present-day climate) and 2091-2100 (future climate, under the RCP8.5 scenario) are analyzed. More

details are provided in Sect. 2 of part 1.





## 3 Methods

We study Lagrangian airstreams in the 1% strongest cyclones in the 10-member CESM-LE dataset for the extended winter season. This cyclone dataset is described in detail in part 1. For the calculation of air parcel trajectories, we apply the Lagrangian Analysis Tool (LAGRANTO; Wernli and Davies, 1997; Sprenger and Wernli, 2015), which uses the three-dimensional CESM-

LE wind field for kinematic forward and backward calculations of air parcel trajectories. 7-day backward trajectories are initialized on a horizontal equidistant grid ($\Delta x = 200$ km) centered at the cyclone center (sea level pressure (SLP) minimum) and covering the storm area with a fixed radius of 1500 km at the time of maximum intensity of each selected extreme cyclone. They are initialized at 8 vertical levels: 850, 800, 700, 600, 500, 400, 300, and 250 hPa. Along the trajectories, several variables are traced, together with the trajectory position (longitude, latitude) and pressure. These physical parameters include specific

humidity $q$, potential temperature $\theta$, and potential vorticity PV. The time axis is defined such that the initialization time (time of maximum cyclone intensity) is $t = 0$, and negative times correspond to periods before the trajectories reach the cyclone.

Composites of Lagrangian tendencies of air mass properties are computed as outlined in the following. The Lagrangian tendency is defined as the change in magnitude of any variable between the initialization time and a previous time step along the backward trajectory for each grid point at a given pressure level and each selected cyclone .These Lagrangian tendencies

are projected on the initialization time step and interpolated on a radial grid centered on the cyclone center, as described for other cyclone-centered composites in section 3.1 of part 1. Finally, the Lagrangian tendencies are averaged over all extreme cyclones to construct the Lagrangian tendency composites.

## 4 Results

The results of this study are presented in three subsections. In order to obtain a general idea of the evolution of key parameters

along the backward trajectories and the relevant time scales, in the first subsection time series are presented pooling all trajectories from specific pressure levels. Subsequently, in the second subsection, the Lagrangian composites on two selected pressure levels are discussed, supplemented by time series of trajectories from individual locations that facilitate linking the composites to the classical airstream concept. Finally, in the third subsection Lagrangian composite cross sections are presented to provide a more comprehensive view on the spatial structure of the Lagrangian transport patterns.

### 4.1 Time series of parameters along trajectories

In this section, we analyze the temporal evolution of different key parameters along the backward trajectories, providing insights into the times during which the parameters experience a significant change, which eventually determines the cyclone intensity and structure. The temporal change of PV and $\theta$ along the trajectories can be associated with diabatic processes due to their conservation for adiabatic and frictionless motion (Hoskins et al., 1985; Madonna et al., 2014, e.g.). A comparison

between the temporal evolution of the present-day and future trajectories may provide insights into the influence of diabatic and advective processes that shape the cyclone structure in the simulated future climate.



As the first step, we evaluate various parameters averaged over all trajectories initialized in the cyclone area, in a radius of $10^o$ around the SLP minimum, at the time of maximum cyclone intensity. In the following, the temporal evolution of a) pressure, latitude, longitude, b) specific humidity, c) $\theta$ and d) PV is discussed for trajectories from two pressure levels, 700 and
250 hPa.

a) Pressure, latitude, longitude

Figs. 2a,b show the temporal evolution of pressure along backward trajectories initialized at 700 and 250 hPa, respectively. Trajectories from both levels indicate ascent in the 24 hours before the air parcels reach the cyclones at the time of maximum
intensity. At low levels, the mean ascent is about 30 hPa, while at upper levels, it is larger, around 50 hPa. This ascent is associated with specific airstreams such as the WCB (Binder et al., 2016; Madonna et al., 2014), as will be discussed in the following subsections. The relatively moderate pressure changes in the mean are due to the fact that, in Fig. 2, data are averaged over the entire cyclone area, leading to partial cancellation between ascending and descending airstreams.

On average, the trajectories experience an equatorward displacement before the ascent and a poleward movement during
the ascent phase (Figs. 2c,d). The temporal evolution of longitude (Figs. 2e,f) shows that the trajectories move eastward. Furthermore, trajectories ending at upper levels travel larger distances, which is due to the typically larger wind velocities at higher altitudes.

In the simulated future climate, the trajectories initialized at 700 hPa are projected to follow a similar zonal, meridional and vertical displacement as in the present-day reference period (Figs. 2a,c,e). In particular, the ascent in the last 24 hours is very
similar, while the mean descent in the phase before these last 24 hours is slightly more pronounced in the future. Trajectories initialized at 250 hPa also follow a similar zonal and meridional path in present-day and future climate (Figs. 2d,f), but origi- nate at lower levels in the future, associated with a larger ascent in the last 24 hours (Fig. 2b).





**Figure 2.** Temporal evolution of (a,b) pressure, (c,d) latitude, (e,f) longitude, (g,h) specific humidity, (i,j) potential temperature and (k,l) PV averaged over all trajectories initialized within a 10° radius around the cyclone center of all selected cyclones and at (left column) 700 hPa and (right column) 250 hPa. The average for present-day climate is shown as blue, dashed line, the average over the future time slice as red line. Shading shows the 5. and 95. percentiles.

b) Specific humidity

Figs. 2g,h show the temporal evolution of specific humidity along the backward trajectories. There are marked differences between the moisture evolution along trajectories reaching the 700 hPa and 250 hPa levels. During the 6-day pre-ascent phase, trajectories reaching the 700 hPa level typically gain moisture. In present-day climate the average specific humidity increases from 1.5 to more than 2 g kg$^{-1}$. During the ascent phase, water vapor condenses, and the specific humidity decreases to 1.5





g kg$^{-1}$. In contrast, trajectories reaching the upper levels experience a constant decrease of moisture with values only slightly
above 0 g kg$^{-1}$ at 250 hPa.

In the future climate, the moisture content increases for both trajectories initialized at upper and lower levels. The moisture
increase for trajectories reaching the cyclones at 700 hPa is constant over time (0.5 gKg$^{-1}$ in the mean). However, trajectories
initialized at 250 hPa converge towards 0 g kg$^{-1}$ when they reach the cyclones, similar to present-day. Accordingly, the mois-
ture decrease in the ascent phase is larger in future than in present-day climate for the upper-level trajectories.

### c) Potential temperature

Figs. 2i,j show an increase of the air parcels' potential temperature during the ascent period ($t = -24$ to $t = 0$) for trajectories
reaching both the 700 and 250 hPa levels. This diabatic heating is linked to the latent heat release associated with the phase
changes of moisture during ascent (condensation, vapor deposition, freezing), in accordance with the decrease of water vapor
specific humidity discussed before. The trajectories reach the level of 700 hPa with a potential temperature of 285 K in the
mean, while the mean $\theta$ is 321 K for trajectories reaching 250. However, there is a large spread of around 20 K.

In the future, the potential temperature along backward trajectories generally increases in the 7 days before maximum cy-
clone intensity. An increase of around 5 K is projected for both trajectories ending at 700 hPa and at 250 hPa. Note that this
increase corresponds to the global mean warming of 5 K. In line with the specific humidity changes, the heating during the
ascent phase in the last 24 hours is similar in future and present-day climate for lower-level trajectories, but there is an amplified
heating in the future (corresponding to the amplified moisture loss) along the 250 hPa trajectories. This is examined in more
detail in the following subsection.

### d) Potential vorticity

Figs. 2k,l show the temporal evolution of PV along the backward trajectories. There are again structural differences between
trajectories initialized at 700 hPa and 250 hPa. Trajectories reaching the cyclones at 700 hPa experience a clear PV increase
in the 24 h before. Such an increase due to diabatic processes is expected for trajetories below the level of maximum diabatic
heating (Binder et al., 2016; Madonna et al., 2014). The mean PV along the 250 hPa trajectories, in contrast, stays relatively
constant. Most likely, these small mean changes are related to opposing PV tendencies along individual trajectories, with some
of them experiencing PV destruction due to their location above the region of maximum diabatic heating. In addition, radiative
cooling and heating may affect the upper-level PV tendencies (Chagnon et al., 2013; Cavallo and Hakim, 2013).

In the simulated future climate, the PV increase along the 700 hPa trajectories is more pronounced than for present-day.
Although there is no indication of an amplified heating in these air parcels (see again Fig. 2i), the heating is likely enhanced
at higher altitudes (see section 4.2 below), as indicated also by an increase in surface precipitation (not shown). The more
pronounced PV increase is thus likely related to a larger vertical gradient in diabatic heating in the future. On the contrary,
the trajectories ending at 250 hPa are projected to have an almost constant mean PV, similar to the present-day trajectories.
Nevertheless, the PV of the future trajectories remains lower than for present-day during the entire 7-day period. This can be



explained by a decrease in the climatological PV at upper levels (see Supplementary Fig. S1). The impact of this PV decrease on the upper PV anomalies differs between cyclone subregions, as demonstrated in more detail in the following.

## 4.2 Lagrangian tendency composites

In the previous section, we have identified that the last 24 hours before the air parcels arrive in the cyclone area at the time of maximum cyclone intensity are a period of particular changes in the physical parameters along the backward trajectories. During this period, both air parcels arriving at 700 and 250 hPa typically ascend, accompanied by latent heat release and, for the lower-level trajectories, a PV increase. However, as demonstrated in part 1, the spatial distribution of PV anomalies as well as its projected future changes vary across the cyclone area, with important consequences for cyclone dynamics and winds. To link such local PV anomalies to the Lagrangian tendencies outlined above, we analyze Lagrangian tendency composites showing the spatial distributions of changes in physical parameters along the backward trajectories in the last 24 hours (see section 3). In the following, we examine such composites of Lagrangian tendencies in latitude, longitude, pressure, $\theta$ and PV.

To facilitate the interpretation of these Lagrangian composites, we also show time series averaged over trajectories that arrive at specific locations in the cyclone composite, namely at the cyclone center as well as $8^o$ to the north and $5^o$ south of the center. Note that these locations correspond to strongly positive (center and northward point), respectively strongly negative (southward point) projected future changes in the upper-level PV anomaly (see Fig. 8c in part 1). The time series thus also help understanding the complex projected upper-level PV anomaly changes in more detail.





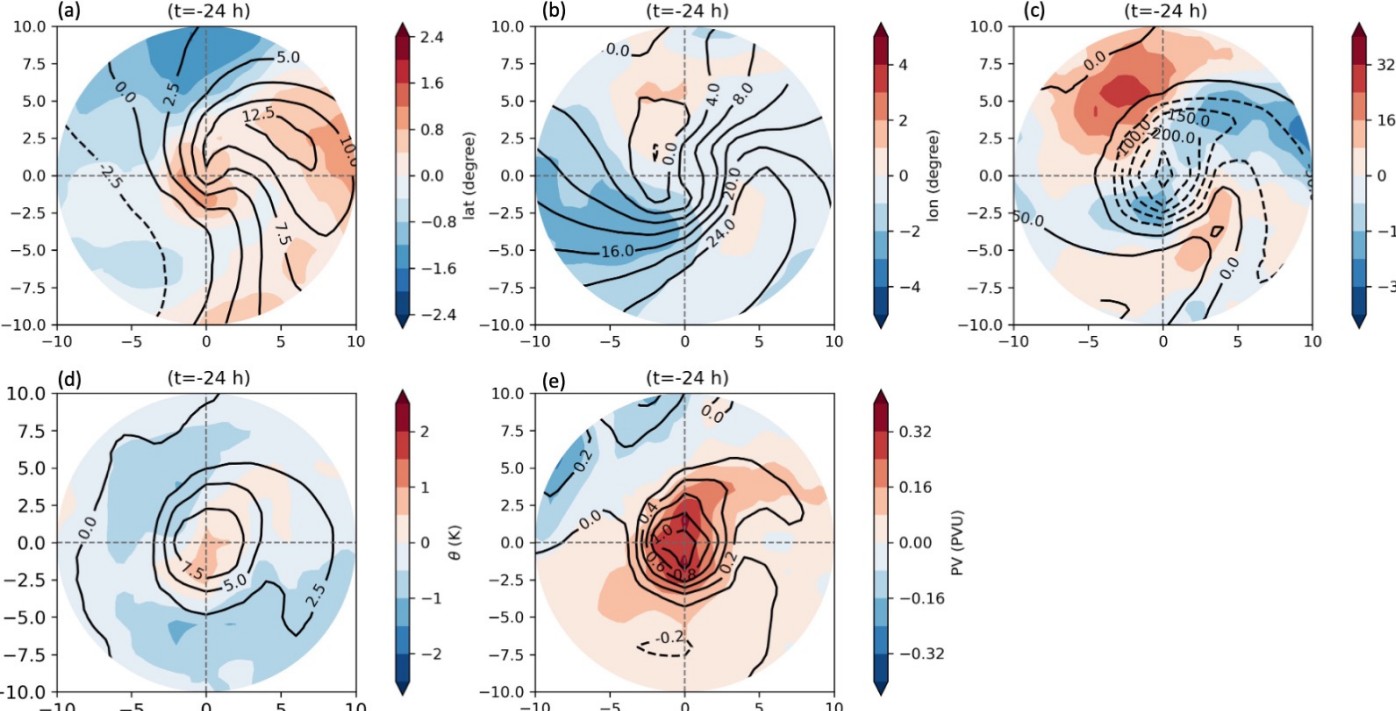

**Figure 3.** Composites of Lagrangian tendencies along backward trajectories initialized at 700 hPa in the last 24 hours before arrival in the cyclone area of (a) latitude, (b) longitude, (c) pressure, (d) potential temperature and (e) PV. Contours show present-day Lagrangian tendencies and the color shading indicates the response to future climate change (difference in the Lagrangian tendencies between future and present-day climate).

### 4.2.1 Lower levels (700 hPa)

Fig. 3a shows the composite of latitude tendencies along trajectories reaching the cyclone area at 700 hPa during the 24 h before the cyclone maximum intensity. The contours indicate the present-day meridional displacement, color shading shows the projected future change in this displacement. Positive values indicate trajectories originating from the south, negative values trajectories from the north. Trajectories from the south dominate the region downstream of the cyclone, with the maximum northward transport near the mean position of the warm front. Trajectories from the north are found southwest of the cyclone

center, with smaller absolute meridional displacements compared to the southerly flow downstream. The composite of longitude tendencies (Fig. 3b) shows that most trajectories move eastward (positive longitude tendency). The largest eastward displacement is found in the warm sector to the south and southeast of the cyclone center. A small region of westward displacement is located northwest of the cyclone center, in the area of the occlusion, indicative of cyclonic wrap-up of the air around the cyclone center. In Fig. 3c, a clear zone of ascent is identified in the warm sector and particularly around the cyclone

center. A maximum rate of ascent of 250 hPa in 24 h, averaged over all trajectories, is found at the cyclone center, whereas





a wedge-shaped region of descent is located to the southwest of the cyclone center. The northward, ascending flow in the cyclones' warm sector can be linked to WCB activity. The descending branch from northwest towards the south of the cyclone center is related to the dry intrusion, as also shown exemplarily in the time series along backward trajectories initialized south of the cyclone center shown in Fig. 4 (left column). These trajectories show the southward transport and descent (particularly

in the last 24 h) of relatively dry air masses from mid-levels. The linkages to the classical air stream concept will be further discussed in section 4.3.

**Figure 4.** Temporal evolution of (a,b) pressure, (c,d) latitude, (e,f) longitude, (g,h) specific humidity, (i,j) potential temperature and (k,l) PV averages along backward trajectories initialized $5^o$ south of the cyclone center and at (left column) 700 hPa and (right column) 250 hPa.



The ascent in the warm sector and near the cyclone center is associated with diabatic heating, as shown by the positive potential temperature tendencies in the same region (Fig. 3d), most probably due to latent heat release during cloud formation. This is further illustrated by the time series initialized in the cyclone center shown in Fig. 5 (left column). In the last 24 hours, these air parcels experience a strong decrease in water vapor specific humidity along with the ascent, northward transport and diabatic heating. The Lagrangian tendency of $\theta$ is also positive in most other parts of the cyclone composite, except for the western boundary (see again Fig. 3d).

Lagrangian PV tendencies are positive in the northern and eastern part of the cyclone area, with the largest PV production at the cyclone center and extending towards the warm front (Fig. 3e). The spatial pattern is similar to the pattern of ascent and diabatic heating. As mentioned above, diabatic PV production due to latent heat release in clouds is expected in these low-level air parcels since they are likely located below the vertical maximum of heat release. A region of PV reduction is found to the south of the cyclone center, albeit with much smaller magnitude compared to the PV generation in other regions. The largest reduction occurs near the expected region of the DI (see also the example in Fig. 4k), which might be related to turbulent mixing (Attinger et al., 2021).





**Figure 5.** Temporal evolution of (a,b) pressure, (c,d) latitude, (e,f) longitude, (g,h) specific humidity, (i,j) potential temperature and (k,l) PV averages along backward trajectories initialized at the cyclone center and at (left column) 700 hPa and (right column) 250 hPa.

Projected future changes of the Lagrangian composites at 700 hPa are shown with color shading in Fig. 3. In the cyclones' warm sector, both the northward motion and ascent are projected to intensify, in particular near the warm front and south of the cyclone center (Fig. 3a,c). Also, the descent in the southern part of the cyclone area intensifies, and on the northwestern flank, enhanced descent is projected in a region of small mean vertical displacement in present-day climate. The mean westerly flow weakens slightly in most regions, in particular southwest of the cyclone center. The spatial pattern of future changes in diabatic heating (Fig. 3d) corresponds relatively well to the change in vertical motion, with regions of stronger ascent experiencing more heating and vice versa. In addition to the enhanced ascent, the larger atmospheric moisture content in a warmer climate explains



the increase in diabatic heating through increasing latent heat release, as shown exemplarily for the cyclone center in Fig. 5g. Finally, Fig. 3e shows that the PV generation in the last 24 hours before maximum cyclone intensity increases in most of the cyclone area, except for the northwestern part. The strongest intensification is found at the cyclone center, extending along the 255 warm front and towards the southwest. This also corresponds to the region of enhanced ascent and diabatic heating, but the relative changes in PV generation are even larger and spatially more extended than the increase in heating (see also Fig. 5i,k). As already discussed in section 4.1, this amplified PV generation can be explained by the increased latent heat release also at higher altitudes (see Supplementary Fig. S2 for the composite change at 500 hPa), leading to a larger vertical heating gradient. The region of an increasing Lagrangian PV tendency in a warmer climate also extends into the southern sector of the cyclone 260 composite where air parcels, on average, descend and the Lagrangian PV tendency is negative in present day climate. This thus indicates a reduced PV destruction in this region (see also Fig. 4k). It may point towards a role of other processes beyond latent heating, in particular turbulent mixing, for future changes in near-surface PV anomalies. Nevertheless, the magnitude of these PV tendency changes is much smaller than the changes near the cyclone center induced by latent heating.

In summary, the most apparent future changes in PV generation occur around the cyclone center. These PV generation 265 changes in the last 24 hours explain a substantial part of the total future composite changes in the cyclones' low-level PV anomalies, also in a quantitative manner (see Fig. 8a in part 1). Our Lagrangian analysis thus provides strong evidence that these PV anomaly changes can be explained by amplified latent heating, driven by both a larger moisture content and enhanced ascent in a warmer climate. This thus corroborates the first part of our hypothesis formulated in section 1. Changes in PV generation further away from the cyclone center are smaller and thus of minor importance for the cyclones' dynamics, as also 270 illustrated exemplarily in the time series along trajectories initialized south and north of the cyclone center shown in Figs. 4 and 6 (left columns), respectively.





**Figure 6.** Temporal evolution of (a,b) pressure, (c,d) latitude, (e,f) longitude, (g,h) specific humidity, (i,j) potential temperature and (k,l) PV averages along backward trajectories initialized $8^o$ north of the cyclone center and at (left column) 700 hPa and (right column) 250 hPa.

### 4.2.2 Upper levels (250 hPa)

The Lagrangian tendency composites at 250 hPa are shown in Fig. 7. In present-day climate, most of the cyclone area is dominated by trajectories coming from the south (Fig. 7a) and west (Fig. 7b). The largest northward displacement occurs northeast of the cyclone center. Near the western boundary of the composite there is a small region with northerly flow, located above a more extended area with northerlies at lower levels (see again Fig. 3a). The maximum eastward displacement southeast of the cyclone center (with a longitude change of more than $45^o$ in 24 h) is co-located with the exit region of an upper-level jet





streak (see Fig. 8d in part 1). The spatial pattern of the regions of ascent and descent shown in Fig. 7c is indicative of cyclonic
wrap-up. Ascending trajectories are located northeast of the cyclone center, with the strongest ascent in the north, which is in

contrast to the pattern at 700 hPa (see again Fig. 3c). These poleward ascending trajectories are related to the WCB outflow, as
also shown in the time series in Fig. 6 (righ column) and discussed further in section 4.3. The descent of trajectories located to
the southwest of the cyclone center is weaker than the ascent in the northeastern part. Similar to th composite at 700 hPa, also
at upper levels the spatial pattern of the Lagrangian tendencies of $\theta$ (Fig. 7d) is similar to the pattern of vertical displacement
(Fig. 3c). Trajectories arriving over the northern and eastern part of the composite region experience diabatic heating, with the

strongest $\theta$ increase in the area of largest ascent, the WCB outflow region. As exemplarily shown in Fig. 6b,h,j, this ascent
from the middle to the upper troposphere goes along with a decrease in specific humidity, indicating that the diabatic heating,
as for the lower level, is most probably due to latent heat release during cloud formation. Descending trajectories southwest of
the cyclone center experience a slight decrease of $\theta$, likely because of longwave radiative cooling.

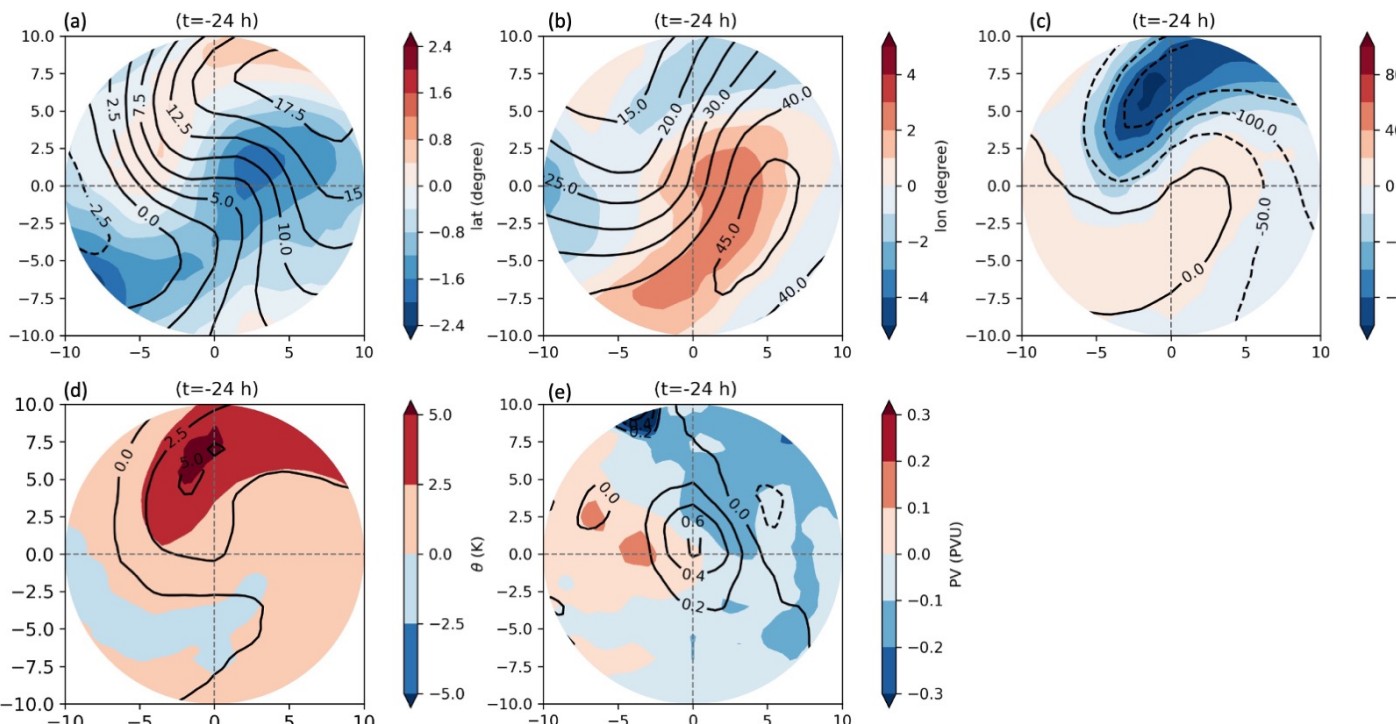

**Figure 7.** Composites of Lagrangian tendencies along backward trajectories initialized at 250 hPa in the last 24 hours before arrival in
the cyclone area of (a) latitude, (b) longitude, (c) pressure, (d) potential temperature and (e) PV. Contours show present-day Lagrangian
tendencies and the color shading indicates the response to future climate change (difference in the Lagrangian tendencies between future and
present-day climate).

The Lagrangian PV tendency in the 24 h before the trajectories reach the cyclone area is shown in Fig. 7e. In contrast to the

lower levels (cf. Fig. 3), here the spatial pattern of the PV tendencies does not correspond to the pattern of heating and vertical





motion. The strongest PV generation occurs in trajectories reaching the cyclone center. As shown in the time series in Fig. 5 (right column), these trajectories are located in the lower stratosphere (PV > 6 PVU) and are very dry (specific humidity < 0.1 g Kg$^{-1}$). The modest increase in $\theta$ (Fig. 5j) and the diabatic PV generation (Fig. 5l) can thus not be explained by latent heat release. As shown by Chagnon et al. (2013) and Cavallo and Hakim (2013), such lower-stratospheric positive PV anomalies can

be diabatically produced by radiative processes near the lowered tropopause above the cyclone center. In particular, longwave radiative cooling is expected to peak in the tropopause region, where vertical humidity gradients are large, leading to a positive vertical heating gradient and thus PV generation above. Farther away from the cyclone center, the mean PV tendency in the last 24 h is close to zero (see again Fig. 7e). This is also the case in the WCB outflow region north of the cyclone center, in spite of substantial latent heat release in these ascending air parcels (see also Fig. 6j,l). Most likely, this is related to varying

positions of the air parcels relative to the spatial maximum of diabatic heating, leading to PV generation in some of them, or in the earlier period of the ascent, and PV destruction in others (those located above the heating maximum), which compensate each other.

Projected future changes in the Lagrangian tendencies at 250 hPa (color shading in Fig. 7) indicate a strengthening of the ascent as well as enhanced northward and reduced eastward transport in the WCB outflow region north of the cyclone

center. This corresponds to an intensification of the WCB outflow that wraps around the cyclone center. This enhanced ascent and WCB activity goes along with amplified diabatic heating (Fig. 7d). The region south of the cyclone center experiences decreasing northward and enhanced easteward transport, which is, however, not related to pronounced changes in vertical motion or diabatic heating. Finally, there are no pronounced future changes in Lagrangian PV tendencies in the last 24 h before the cyclones' maximum intensity (Fig. 7e). The PV tendencies slightly decrease northeast of the cyclone center, but this

decrease is small compared to the projected change in PV advection in this region (see example in Fig. 6l).

Hence, again, in contrast to the 700 hPa level, the 24 h Lagrangian PV tendency changes do not explain the projected changes in PV anomalies at 250 hPa (Fig. 8c in part 1). The projected PV anomaly reduction south of the cyclone center can be explained by a strong PV reduction at the origin of the air parcels seven days before arriving in the cyclone area (Fig. 4l). These lower PV values at the origin, which are approximately conserved during transport (at least in the mean over all trajectories), are due to a

future reduction in the mean PV climatology (Supplementary Fig. S1), but also a mean downward and, in particular, southward shift in the location of the origin (Figs. 4b,d). The reduced PV anomalies in this southern region are thus primarily caused by enhanced meridional transport of low-PV air from more southerly regions, which is consistent with the mean wind changes shown in Fig. 8d of part 1. Also for trajectories arriving at the cyclone center, the PV at their origin seven days before is slightly lower in the simulated future climate (Fig. 5l), but this difference is compensated mostly by enhanced PV production in the

last 24 h. As discussed above, this diabatic PV change is likely associated with radiative processes. The trajectories thus arrive at the cyclone center with almost the same PV in present-day and future climate, but, because of the decrease in climatological PV (Supplementary Fig. S1), this leads to a larger positive PV anomaly in the future climate, explaining the positive signal at the cyclone center in Fig. 8c of part 1. Finally, the projected increase of PV anomalies northeast of the cyclone center (see again Fig. 8c in part 1), which cannot be explained by our Lagrangian diagnostic (Fig. 6, right column), should be interpreted

with care, as it is not robust across vertical levels (in a region of large vertical PV gradients) and (at least at many grid points)





ensemble members. An increase in PV anomalies in the northeast is thus not apparent in the upper-tropospheric vertical mean change in Fig. 9b of part 1, and is thus not thought to substantially affect future changes in cyclone dynamics.

In summary, several processes shape future changes in cyclonic PV anomalies at upper levels, in particular, changes in PV advection, but possibly also radiative processes in the lower stratosphere near the cyclone center. The latter should be investi-
gated in more detail in future studies. Amplified latent heat release affects the WCB outflow region and is linked to enhanced vertical motion there, but this does not go along with consistent changes in PV anomalies, likely due to a compensation between a strengthening of both diabatic PV generation and reduction. All together, this corroborates the second part of our hypothesis formulated in section 1.

### 4.3 Lagrangian composite cross sections

The Lagrangian composite analysis has shown that transport and diabatic processes along trajectories shape future changes in cyclonic PV anomalies on specific pressure levels. To better understand the three-dimensional structure of such Lagrangian tendency changes and further link these tendencies to the classical airstream concept (see section 1), in the following, this analysis is extended with the help of vertical cross sections of Lagrangian pressure and PV tendencies. More specifically, four cross sections are studied: (1) west to east through the cyclone center, (2) south to north through the cyclone center, (3) south
to north $8^o$ east of the center (WCB region), and (4) west to east $4^o$ south of the center (dry intrusion region).

### 4.3.1 Cyclone center cross sections

Figure 8 shows the two cross-sections (1) and (2), which are located at the center of the composite cyclone. The west-east section indicates the presence of two regions of strong ascent in present-day climate (Fig. 8a). The first region is located above the cyclone center, with the most vigorous ascent at mid-levels. This region extends further to the north of the center and to
higher altitudes, as shown in Fig. 8c. The second region is located in the eastern part of the cross section, with the strongest ascent around 400 hPa. Both the ascent at high levels north of the cyclone center and the ascent in the eastern part of the cross section are likely associated with WCBs, more specifically with a cyclonic WCB branch wrapping around the cyclone center and an anticyclonic branch extending further downstream, respectively, as already described in previous studies (Martínez-Alvarado et al., 2014; Binder et al., 2016). An area of descent is located to the south of the cyclone center, peaking around
700-650 hPa (Fig. 8c), which is probably linked to the dry intrusion.





**Figure 8.** (a,b) West-east and (c,d) south-north vertical cross sections through the cyclone center. Contours show present-day Lagrangian tendencies along backward trajectories in the last 24 hours before arrival in the cyclone area of (a,c) pressure and (b,d) PV. Color shading shows the response to future climate change.





The strongest PV production occurs along trajectories initialized near the cyclone center at low levels (Figs. 8b,d), with a maximum PV increase of more than 1.2 PVU in 24 h. It extends up to mid-levels. This PV production, as described above, is due to latent heat release during cloud formation. A second maximum in PV production is located above the cyclone center at upper levels, peaking around 300 hPa, above the region of strong ascent. As discussed in section 4.2.2, the diabatic PV
generation in this region near and above the tropopause is likely due to radiative heating and cooling (Chagnon et al., 2013; Cavallo and Hakim, 2013). Together, the PV generation by these diabatic processes at lower and upper levels contributes to the formation of a PV tower above the cyclone center. The PV tendencies in the air parcels associated with the WCB are positive in the mid-level region of strong ascent above the cyclone center, contributing to cyclone intensification (see again Binder et al., 2016), and small to slightly negative in the upper-level outflow regions to the north and east of the cyclone center. The
latter, as discussed above, is due to the compensation between PV generation below and PV destruction above the heating maximum (Madonna et al., 2014; Methven, 2015). In spite of these small Lagrangian PV tendencies, WCB outflow typically leads to negative PV anomalies in the upper troposphere due to net upward transport of low-PV air from lower levels (Steinfeld and Pfahl, 2019; Grams et al., 2011). In the region of mean descent south of the cyclone center, Lagrangian PV tendencies are negative (see also the example in Fig. 4k), possibly associated with turbulent mixing of the descending air parcels with
lower-tropospheric, low-PV air (Attinger et al., 2021).

In the simulated future climate, the ascent in the cyclones' core region near the center is projected to intensify over the entire column (blue shading in Figs. 8a,c), with a maximum intensification at upper levels slightly west of the center that corresponds to an upward extension of the region of strong ascent in the cyclonic branch of the WCB outflow. This enhanced ascent goes along with amplified PV production at low- and mid-levels due to stronger latent heat release (red shading in Figs. 8b,d). Two
regions are identified where the PV tendency increases by more than 0.2 PVU: the lower levels above the cyclone center (see also Fig. 5, left column) and the core region of the WCB ascent between 600 and 500 hPa slightly west of the center. On the contrary, there is no clear future change in the radiative PV generation at upper levels above the cyclone center. The increase observed at 250 hPa in Fig. 5l is thus not representative for the altitudes slightly below. This apparently complex response of such radiative processes near the tropopause to climate warming warrants future investigation. In the upper-tropospheric
outflow region of the cyclonic branch of the WCB north of the cyclone center, the ascent intensifies above 300 hPa and also shifts upwards (dipole pattern in Fig. 8c). The changes in PV tendencies in this region are relatively small, but consistently negative in the upper part (300 hPa and above). Such a reduction of PV generation may be partly explained by the upward shift of the tropopause and WCB outflow in a warmer climate, leading to weaker PV tendencies in a region characterized by radiative PV production in the lower stratosphere in present-day climate. In the anticyclonic branch of the WCB in the eastern
part of the cross section, the ascent is projected to weaken at mid- and upper-levels, without a clear effect on PV tendencies. Finally, the region of descent south of the cyclone center shifts upward, associated with intensified descent at mid-levels. PV tendency changes in this area are again small, but positive, indicating a weakening of the PV reduction in the descending air parcels. In the next two subsections, two additional cross sections are analyzed to better characterize the changes in the eastern WCB branch as well as the dry intrusion region of descent south of the center.



### 4.3.2 WCB cross section

As described above, we have identified two WCB branches. The evolution of the first, cyclonic branch has been described in detail with the vertical cross sections at the cyclone center. To analyze the second branch, we have constructed the south-north cross-section (3), $8^o$ east of the center, as shown in Fig. 9.

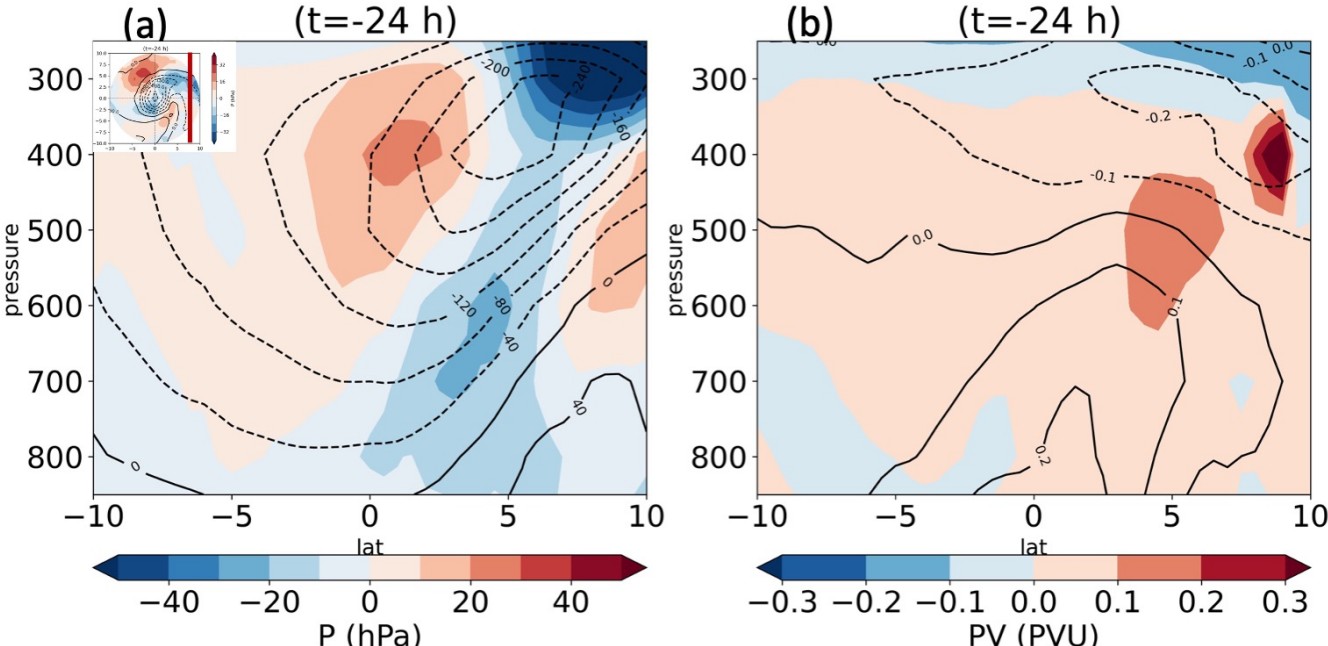

**Figure 9.** South-north vertical cross section through the WCB region $8^o$ east of the cyclone center. Contours show present-day Lagrangian tendencies along backward trajectories in the last 24 hours before arrival in the cyclone area of (a) pressure and (b) PV. Color shading shows the response to future climate change.

The cross section is dominated by ascending trajectories (Fig. 9a), with the strongest ascent of more than 240 hPa in 24 h at upper levels in the northern part associated with the WCB outflow. A dipole pattern is apparent in the Lagrangian PV tendencies (Fig. 9b), with PV generation below 500 hPa (albeit with a smaller magnitude compared to the cyclone center, cf. Fig. 8b,d) and PV destruction above, in particular in the northern part. This large area of negative PV tendencies may contribute to ridge building downstream of the cyclones (see again Steinfeld and Pfahl, 2019; Grams et al., 2011).

In future climate, the ascent of this WCB branch is projected to shift poleward, associated with reduced ascent on the southern flank of the present-day maximum (peaking at 400 hPa) and enhanced ascent on its northern flank (throughout the entire column). The strongest increase in ascent is found above 300 hPa near the northern edge of the cross section, implying an upward extension of the WCB outflow, similar to the cyclonic branch. Figure 9a also shows that the reduced ascent observed for the eastern WCB branch in Fig. 8a does not imply a weakening of the entire branch, but, as discussed above, rather a spatial shift. PV tendencies are projected to increase across most of the vertical section from the surface up to 300 hPa (Fig. 9b). In





some areas around 400 hPa, this changes the sign of the tendency, from PV destruction in present-day to PV generation in future climate. The strongest PV tendency increase is found above the area of the strongest mid-level amplification of ascent, which is consistent with enhanced PV generation by latent heat release in the ascending air masses. It also coincides with increasing precipitation in the region of the warm front (not shown).

### 4.3.3  DI cross section

The Lagrangian tendency composite at 700 hPa, the north-south vertical cross-section through the cyclone center and the time series in Fig. 4 (left column) show a descending airstream south of the center with characteristics of a dry intrusion (Raveh-Rubin, 2017). To study the spatial pattern of this DI in more detail, we analyze the west-east vertical cross-section (4) 4$^o$ south of the cyclone center, as shown in Fig. 10.

In this cross section, there is a broad region of descent in the western part and ascent in the eastern part (Fig. 10a), the

latter being associated with the WCB. The transition between these two regimes around 5$^o$ east of the cyclone center is likely associated with the mean position of the cold front. The two areas of strongest descent are located at the western boundary at low levels (∼800 hPa) and east of the cyclone center at middle levels (∼550 hPa). Apparently, some DI trajectories arrive to the west of the cyclone moving southeastward at low levels and others to the east of the cyclone, moving northeastward close to the cyclone center. This is consistent with the fanning out of DI trajectories as they descend behind the cold front described

in previous studies (Browning, 1997). Lagrangian PV tendencies are negative in the regions of strongest descent and positive below the maximum descent east of the cyclone center (Fig. 10b).





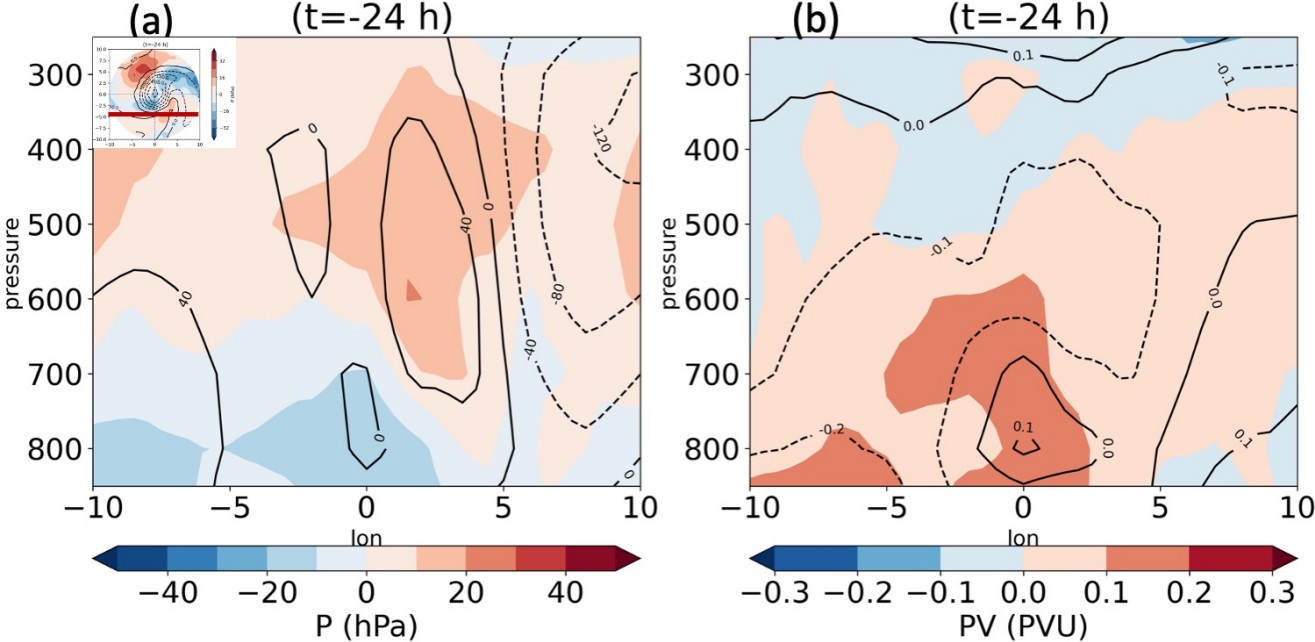

**Figure 10.** West-east vertical cross section through the dry intrusion region $4^o$ south of the cyclone center. Contours show present-day Lagrangian tendencies along backward trajectories in the last 24 hours before arrival in the cyclone area of (a) pressure and (b) PV. Color shading shows the response to future climate change.

In a warmer climate, the descent of the DI trajectories at low levels in the western part is projected to weaken. In contrast, at the mid-level maximum east of the cyclone center, the descent is amplified. PV tendencies increase in most areas below 500 hPa, weakening the PV destruction in most regions, but amplifying PV generation below the area of maximum descent south of the cyclone center. These projected changes in DI descent and PV tendencies may have implications for severe weather events associated with the DI, which should be investigated in more detail in future studies.

# 5 Conclusions

In this study, we have used a Lagrangian perspective based on air parcel trajectory calculations to investigate projected future changes of air streams in intense North Atlantic extratropical winter cyclones as well as their role for PV anomaly changes. Lagrangian tendency composites have been constructed to link this Lagrangian perspective to the Eulerian composite analysis of part 1 of this study.

Our results show that lower-tropospheric air parcel properties, in particular PV, are driven by changes happening in the last 24 hours before the air parcels arrive in the cyclone area. This period is strongly affected by diabatic heating associated with cloud formation in ascending air streams, which leads to diabatic PV production maximizing at and around the center of the composite cyclone. Primarily due to the increasing atmospheric moisture content in a warming climate, this diabatic



PV generation on the day before the cyclones' maximum intensity is projected to intensify, which quantitatively explains the increase in low-level cyclonic PV anomalies that has been identified in part 1 and that has a strong impact also on low-level wind changes, in particular, in the cyclones' warm sector. The amplified latent heating is also associated with enhanced ascent and PV generation in WCB airstreams (in agreement with previous studies based on the same CESM simulations, see Joos

et al., 2023; Binder et al., 2023), which further contributes to the intensification of midlevel PV anomalies in the cyclones' PV tower. Accordingly, our Lagrangian analysis corroborates the hypothesis that the relative importance of moist processes, especially latent heat release during cloud formation, for cyclone dynamics increases in a warmer climate compared to other, dry-dynamical mechanisms, which has already been formulated in part 1 and other previous studies (Pfahl et al., 2015; Büeler and Pfahl, 2019; Sinclair et al., 2020; Binder et al., 2023). Changes in low-level PV tendencies and anomalies in regions further

away from the cyclone center, including the DI region to the south, are of smaller magnitude and thus likely of secondary importance for general aspects of the cyclones' dynamics. Nevertheless, projected changes in DI descent and diabatic PV destruction (likely associated with turbulence) may have implications for more localized, smaller-scale severe weather events that should be investigated in future studies.

Future changes in air parcel properties at upper-level are more complex and affected by several processes also operating

on time scales longer than 24 hours. Accordingly, the upper-level composite PV anomaly changes identified in part 1 do not correspond to the 24 h Lagrangian PV change composite shown in Fig. 7e). Enhanced vertical and especially meridional transport of low-PV air masses explain the decrease of PV anomalies south of the cyclone center. An increase in the stratospheric anomaly above the center is related to a combination of altered transport, diabatic PV generation likely due to radiative processes, and changes in the background PV climatology. In particular the complex changes in radiative PV production near the

tropopause level warrants further investigation. Air parcels arriving in the WCB outflow regions north (cyclonic branch) and northeast (anticyclonic branch) of the cyclone center experience enhanced ascent, leading to an elevated WCB outflow level, and a modest intensification of diabatic PV destruction. However, this is not associated with robust and consistent changes in PV anomalies in the outflow regions, most likely because the enhanced PV destruction is partly compensated by enhanced PV generation during an earlier period, when the air parcels are located at lower altitudes, and also by the complex interplay

of changes in other processes (advection, radiation) and the spatial pattern of the WCB outflow that may mask the relatively small cloud-diabatic tendency changes. Nevertheless, note that the lack of robust changes in PV anomalies on the 250 hPa level does not implicate that the enhanced WCB ascent and diabatic heating may not have an effect on the persistence and intensity (through the elevated outflow level) of downstream ridges and atmospheric blocking (cf. Steinfeld et al., 2022).

Limitations of our study are related to the fact that it is based on a single climate model only. As shown previously, the

CESM simulations applied here capture WCB properties and their connection to cyclones reasonably well, but still underestimate the magnitude of positive low-level PV anomalies associated with the WCB (Joos et al., 2023; Binder et al., 2023). Future investigations should thus focus on the robustness of the conclusions obtained here when using more than one model. Furthermore, the relatively coarse spatial and temporal resolution (time step of 6 hours) limits the ability of the trajectory calculations to resolve ascending motion. This implies that, for instance, convection embedded in the WCB is not captured (Oertel et al.,

2020), and the influence of diabatic changes on future cyclone dynamics may be underestimated (Willison et al., 2015).



*Code and data availability.* The code of the CESM version 1 that was used for the Large Ensemble simulations is available from https://www.cesm.ucar.edu/models/cesm1.0/. The code of the trajectory model LAGRANTO is available from https://iacweb.ethz.ch/staff/sprenger/lagranto/download.html. The model output of the CESM-LENS re-runs used here is available upon request from the authors.

*Author contributions.* S.P. and E.D.-T. designed the study. E.D.-T. performed the analysis, produced the figures and drafted the manuscript.
Both authors discussed the results and edited the manuscript.

*Competing interests.* Stephan Pfahl is executive editor of WCD.

*Acknowledgements.* We are grateful to Urs Beyerle (ETH Zurich) for performing the CESM-LENS re-runs. S.P. acknowledges support by Deutsche Forschungsgemeinschaft through Grant CRC 1114 'Scaling Cascades in Complex Systems', project C06. We acknowledge the HPC service of ZEDAT, Freie Universität Berlin, for providing computational resources (Bennett et al., 2020).



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
