# Peer review of "Future changes in North Atlantic winter cyclones in CESM-LE -Part 2: A Lagrangian analysis"

_EGUsphere, 2023_

## Referee Comment (RC1)

Future changes in North Atlantic winter cyclones in CESM-LE – Part 2: A Lagrangian analysis
egusphere-2023-1382

The aim of this study is to investigate the processes leading to projected future changes in mid- and upper-level PV anomalies through a Lagrangian analysis of cyclone airstreams. The authors analyse changes in several variables along Lagrangian back trajectories initiated at different locations within the cyclone composites.  They conclude that the majority of the PV tendencies occur within the last 24 hours before they reach their initiation point. They attribute the low-level PV tendencies to ascent in the WCB but cannot simply attribute upper-level PV tendencies to a cyclone airstream.  The figures are well presented, and the structure of the paper is easy to follow. I enjoyed reading the paper.

The authors have attempted to link their previous Eulerian analysis to this Lagrangian analysis which is interesting, particularly the cyclone-centred composites of Lagrangian tendencies.  My main concern about the analysis, is that the cyclone airstreams discussed are not explicitly identified.  A cartoon of the airstreams is shown in figure 1, but the same airstreams are not identified with sufficient accuracy in the analysis of the data (see general comments below).  As the aim of the paper is to link PV anomalies to cyclone airstreams, I think this needs to be addressed before the paper is suitable for publication.

Major comments
Figures 3 and 7 show cyclone-centred composites of Lagrangian tendencies and how they are projected to change in the future.  These figures are nicely presented but I struggled to identify the cyclone airstreams in these figures.

On line 226 the authors link the northward, ascending flow in the cyclone's warm sector to the WCB.  The region of maximum ascent is located close to the cyclone centre, but the region of maximum poleward displacement is located further north-east, what region specifically is linked to the WCB and how does this relate to the WCB illustrated in figure 1?  Furthermore, the WCB is typically comprised of two branches, one ascending and turning anticyclonically at upper levels and another ascending and turning cyclonically at mid-levels.  While there is evidence of the anticyclonic branch in figure 7b, there is no evidence of the cyclonically turning branch.  Line 304-305 states that the reduced eastward transport in the WCB outflow region corresponds to an intensification of the WCB outflow that wraps around the cyclone centre, but the flow is still westward and hence not cyclonic. Is this because the cyclonic branch is located at a lower pressure level? If so, can cyclone-centred composites of Lagrangian tendencies at this lower pressure-level be shown. The cyclonic branch is also missing from the figure 1 illustration. Line 347 states that ascent in the eastern part of figure 8c is associated with the cyclonic WCB branch wrapping around the cyclone centre.  Please can the authors present evidence of this cyclonic branch. Finally, line 450 refers to the cyclonic and anticyclonic branches of the WCB. More evidence is needed to support this conclusion.

Line 227 links the descending southward flow to the DI.  Like the WCB, the DI is typically comprised of 2 branches, one turning cyclonically at low-levels and another turning anticyclonically near the surface (as stated on line 74).  The anticyclonic branch is missing from the figure 1 illustration. While there is evidence of the cyclonic branch in figure 3b, there is no evidence of the anticyclonically turning branch. Also, in line 413 the authors state that some DI trajectories arrive to the west of the cyclone moving southeastward at low levels and others to the east of the cyclone moving northeastward close to the cyclone centre. Is this motion shown in figure 10a?  I do not see any eastward motion in this figure, which shows pressure tendencies, or in figure 3b which shows longitudinal tendencies.

The authors state on line 71 that the CCB can produce PV anomalies in the lower and middle troposphere, but analysis of this airstream is entirely missing from the paper. They also state that the CCB consists of 2 branches (line 67) but only the cyclonic branch is shown in figure 1 for some

reason. Is this because no identification of the CCB airstream is possible from the data using the current latitude and longitude tendencies (figures 3a and b).

To address the points above, the authors should also show figures of the cyclone-relative tendencies of the trajectories. I.e., subtract the cyclone motion 24hr latitudinal and longitudinal tendency from the trajectory tendencies. This will illustrate the cyclone relative trajectory tendencies and will likely highlight the missing WCB and DI branches and the CCB.

Minor comments
1. Line 103. Should 'proving' be 'providing'?
2. Line 148. If averaging over the entire cyclone area leads to cancellation between ascending and descending airstreams, why is this analysis presented? They also have a very large spread (line 176) meaning that interpretation of the averages is difficult.
3. Figure 2. Is the shading around the present-day average the grey or red shading?
4. Line 187: In the 24 h before what?
5. Line 220. I suggest that the trajectories from the north have smaller absolute meridional displacement because the cyclone's themselves are typically travelling northwards enhancing to the airstream trajectory component in that direction (see major comments).
6. Line 223. I suggest that the relatively small region of westward displacement would be more significant if cyclone-relative longitudinal tendencies were plotted. This would give a better indication of cyclonic wrap-up of the air around the cyclone centre.
7. Figure 5 and others. I think the description of blue and red lines should also be in the figure caption.
8. Line 281.'Righ' should be 'right'.
9. Line 282. 'th' should be 'the'.

---

## Referee Comment (RC3)

Review of "Future changes in North Atlantic winter cyclones in CESM-LE – Part 2: A Langrangian analysis" by Dolores-Tesillos and Pfahl.

This manuscript uses back trajectories to better understand how the potential vorticity structure and thus the dynamics of extra-tropical cyclones will change in a warmer climate. The main conclusions are that in a warmer and moister climate, enhanced ascent and latent heating in warm conveyor belts leads to a stronger low-level PV anomaly. In contrast, upper-level PV anomalies response in a much more complicated manner which the authors show is due to changes in advection and hypotheses that changes to radiative cooling near the tropopause are also important. Overall, the manuscript is well written, easy to follow, and the conclusions are well supported by the presented evidence. I have two concerns with this manuscript which I describe below, and numerous rather minor comments also listed below.

Major comments:

1. Almost all of the results are presented as averages over all extreme cyclones. Cyclones are highly variable in their structure and dynamics. The impact of this variability is not taken into consideration in this study. Specifically:

   a) Line 137-138 *"we evaluate various parameters averaged over all trajectories initialised in the cyclone area, in a radius of 10 degrees around the SLP minimum"* – this is a huge area and includes air masses with very different properties e.g., the cold sector, the warm convector belt. This huge variability is seen in Figure 2. Does this make scientific sense to average so many different trajectories together? However, in the other extreme, the authors then proceed to show trajectories from just one grid point (Figures 4, 5 and 6) which potentially are not representative. I strongly encourage the authors to re-consider their approach as I expect that much clearer and informative results may be obtained if trajectories only from certain areas of the cyclone were averaged together. Another recommendation, if the approach in Figures 4 – 6 is kept, is to include a measure of uncertainty on these figures, similar to what is done in Figure 2.

   b) How do the magnitudes of the changes detected relate to the amount of variability in the control simulation? Or stated another way, are these results statistically significant? Figures 3 and 7 should include information showing where the changes are significant.

2. Section 3. Some additional details of the simulations should be added here as it is not reasonable to expect a reader to read part 1. Even some basic information such as what time periods the simulations cover (this is in the abstract but could be repeated here), what resolution the simulations are performed at (the coarse resolution is noted as a limitation of this study in the conclusions, but a reader is not told what it is) would be appreciated. I also suggest that a few more details are given about the strongest 1% of cyclones – how many cyclones are there in absolute numbers in both the historical and future climate simulations? Do they all occur in a certain part of the north Atlantic or do they cover a huge geographic area? What metric is used to measure intensity?

Minor comments:

1. Line 52. Units Wm-2 is missing the negative sign.
2. Line 58. "This PV ascent and descent"… This is rather strange, suggest revising it.

3. Line 69 – 70. This second branch of the cold conveyor belt is never mentioned again in the results section / the analysis so does it really exist on average or is this a rare feature?
4. Line 163, these values of specific humidity seem to be very small, however, it may be due to the large area that they are averaged over. Is this a valid hypothesis?
5. Lines 170 – 200. This section discusses many of the processes we would expect in the warm conveyor belt, yet the results being discussed include all of the cyclone areas. This section should at least reminder a reader that the average trajectories also include those arriving in the cold sector.
6. Line 210 – could the location of these points be added to a composite map?
7. Figures 3 and 7. The units on the colour bar on panel (c) are missing. It might also be a good idea to state in the caption here how many cyclones these composites were created from.
8. Line 282 – typo "th" → the
9. Figure 7e. There is a small area of negative PV tendency in the control simulation. I don't think this is discussed in the text. Is this related to negative PV tendencies about the localised heating maximum in the warm conveyor belt?
10. Line 310 – 327. There are many references to figures / results in part 1. This makes it quite difficult for a reader to follow without going to find the figures in part 1. Could this be revised so a readers' understanding does not require part 1?
11. Line 405-406. Can the references to the figures be added here? e.g., figure 2 for the Lagrangian composite at 700hPa.
12. Line 424. Could add here what intense really means e.g. top 1% which is X numbers of cyclones.

---

## Author Comment (AC1)

**Answer to Reviewer 1**

The aim of this study is to investigate the processes leading to projected future changes in mid-and upper-level PV anomalies through a Lagrangian analysis of cyclone airstreams. The authors analyse changes in several variables along Lagrangian back trajectories initiated at different locations within the cyclone composites. They conclude that the majority of the PV tendencies occur within the last 24 hours before they reach their initiation point. They attribute the low-level PV tendencies to ascent in the WCB but cannot simply attribute upper-level PV tendencies to a cyclone airstream. The figures are well presented, and the structure of the paper is easy to follow. I enjoyed reading the paper.

The authors have attempted to link their previous Eulerian analysis to this Lagrangian analysis which is interesting, particularly the cyclone-centred composites of Lagrangian tendencies. My main concern about the analysis, is that the cyclone airstreams discussed are not explicitly identified. A cartoon of the airstreams is shown in figure 1, but the same airstreams are not identified with sufficient accuracy in the analysis of the data (see general comments below). As the aim of the paper is to link PV anomalies to cyclone airstreams, I think this needs to be addressed before the paper is suitable for publication.

We appreciate and thank you for reading our manuscript and giving such constructive feedback. Below, we address your concerns point by point. The figure and line numbers refer to the original manuscript. The reviewer comments are in black and our responses are highlighted in blue.

Major comments

Figures 3 and 7 show cyclone-centred composites of Lagrangian tendencies and how they are projected to change in the future. These figures are nicely presented but I struggled to identify the cyclone airstreams in these figures.

Thanks for this detailed and constructive comment. Our main goal is to link the Eulerian composite changes identified in part I to Lagrangian changes in air mass trajectories and properties, for which, in our opinion, the Lagrangian composites are a useful tool. Nevertheless, as you have emphasized, linking these composites to the classical air stream perspective is complicated, e.g., due to the fact that the composites show averages over many trajectories arriving at the same location relative to the cyclone center (meaning that, for instance, if some trajectories move westward and others eastward, the mean effect will be a small change) and that the tendencies from different locations in the composite do generally not refer to the same air masses (we thus cannot easily trace specific air streams through the composites). We have decided to not try to make the link with the air streams more explicit, e.g., through identifying the air streams with quantitative criteria (such as the 600 hPa ascent criterion for WCBs, see Madonna et al., 2014), because this would have added another angle to an already methodologically complex study, and because future changes in WCBs identified in this way have already been studied in the same model simulations (Joos et al., 2023; Binder et al., 2023). Nevertheless, we think that a qualitative comparison of our Lagrangian composite results with the air stream concept is useful. This also follows previous studies (e.g., Catto et al., 2010; Dacre et al., 2012) that discussed cyclone air streams based on Eulerian composites. We will improve the corresponding discussion in the revised manuscript, in

particular, by adding a new figure showing the Lagrangian lon/lat changes relative to the cyclones displacement, as you have suggested below (see Fig. R1 in this document).

On line 226 the authors link the northward, ascending flow in the cyclone's warm sector to the WCB. The region of maximum ascent is located close to the cyclone centre, but the region of maximum poleward displacement is located further north-east, what region specifically is linked to the WCB and how does this relate to the WCB illustrated in figure 1? Furthermore, the WCB is typically comprised of two branches, one ascending and turning anticyclonically at upper levels and another ascending and turning cyclonically at mid-levels. While there is evidence of the anticyclonic branch in figure 7b, there is no evidence of the cyclonically turning branch. Line 304- 305 states that the reduced eastward transport in the WCB outflow region corresponds to an intensification of the WCB outflow that wraps around the cyclone centre, but the flow is still westward and hence not cyclonic. Is this because the cyclonic branch is located at a lower pressure level? If so, can cyclone-centred composites of Lagrangian tendencies at this lower pressure-level be shown. The cyclonic branch is also missing from the figure 1 illustration. Line 347 states that ascent in the eastern part of figure 8c is associated with the cyclonic WCB branch wrapping around the cyclone centre. Please can the authors present evidence of this cyclonic branch. Finally, line 450 refers to the cyclonic and anticyclonic branches of the WCB. More evidence is needed to support this conclusion.

The absence of spatial alignment of the regions of maximum ascent and maximum poleward transport in the composites at 700 hPa is related to the fact that different locations in the composites air associated with different air masses. The air parcels near the cyclone center appear to have ascended most before arriving at 700 hPa, while the air parcels in the warm sector east of the center have experienced a stronger northward displacement, but slightly less ascent. Nevertheless, these air parcels in the warm sector are embedded in a vertically extended region with strong ascent (maximum 24h ascent at upper levels, see Fig. 9a), which is most likely a signature of the warm conveyor belt. This is what we mean in line 226, and we will try to make this more explicit in the revised manuscript.
To identify the cyclonic WCB branch, the new Fig. R1 is particularly useful, as anticipated by the reviewer. While there is still no mean westward flow relative to the cyclone center in the composite at 250 hPa, a region of westward motion is evident at 500 hPa, indicating that the outflow of the cyclonic branch is located at somewhat lower altitudes, as suggested by the reviewer. This is consistent with the region of maximum ascent in the cross section in Fig. 8a and will be discussed in more detail in the revised manuscript. The cyclonic branch will also be added to the schematic illustration in Fig. 1.

Line 227 links the descending southward flow to the DI. Like the WCB, the DI is typically comprised of 2 branches, one turning cyclonically at low-levels and another turning anticyclonically near the surface (as stated on line 74). The anticyclonic branch is missing from the figure 1 illustration. While there is evidence of the cyclonic branch in figure 3b, there is no evidence of the anticyclonically turning branch. Also, in line 413 the authors state that some DI trajectories arrive to the west of the cyclone moving southeastward at low levels and others to the east of the cyclone moving northeastward close to the cyclone centre. Is this motion shown in figure 10a? I do not see any eastward motion in this figure, which shows pressure tendencies, or in figure 3b which shows longitudinal tendencies.

There is no indication of an anticyclonic DI branch in our composites. This could be associated with case-to-case variability in the occurrence and location of this air stream, which leads to cancellation effects in the composites. For instance, previous studies have shown that the anticyclonic branch can be located relatively far away from the sea level pressure minimum (Catto et al., 2010; Fluck and Raveh-Rubin, 2023). With regard to the second part of the comment, we are not sure what the reviewer refers to, as there is prevalent eastward motion 5° south of the cyclone center in Fig. 3b (and also the new Fig. R1 throughout the troposphere).

Note also that some structural differences can be found between our results and Dacre et al., 2012, with regard to the location of the DI region. In our case, the descending trajectories are located south of the cyclone center instead of upstream. These differences can be attributed to the fact that we do not rotate the fields in the storm direction.

For consistency with previous studies, we will add an anticyclonic DI branch in Fig. 1 and add a short comment to the manuscript that this branch, however, cannot be identified in our results.

The authors state on line 71 that the CCB can produce PV anomalies in the lower and middle troposphere, but analysis of this airstream is entirely missing from the paper. They also state that the CCB consists of 2 branches (line 67) but only the cyclonic branch is shown in figure 1 for some reason. Is this because no identification of the CCB airstream is possible from the data using the current latitude and longitude tendencies (figures 3a and b).

As for the DI, we will modify Fig. 1 to show both branches of the CCB. However, the CCB is not evident in our Lagrangian composites and thus not further discussed in the manuscript. We will add a corresponding note to the introduction.

To address the points above, the authors should also show figures of the cyclone-relative tendencies of the trajectories. I.e., subtract the cyclone motion 24hr latitudinal and longitudinal tendency from the trajectory tendencies. This will illustrate the cyclone-relative trajectory tendencies and will likely highlight the missing WCB and DI branches and the CCB.

Following your suggestion, we will add a figure with the cyclone-relative tendencies (Fig. R1) to the manuscript. At low levels, in present-day (contours), the Figure provides more comprehensive evidence of the trajectories traveling to the south upstream and north downstream of the cyclone. Westward trajectories are more evident to the northwest of the cyclone center, while eastward trajectories are more evident to the southeast of the cyclone center. Thus, we confirmed the WCB location, ascending to the southeast of the cyclone center and wrapping up northwest of the cyclone center at middle levels.

Note that the future changes in cyclone-relative tendencies are very similar to the absolute tendencies shown in Figs. 3 and 7 in the manuscript.

[Figure]

Figure R1. Composites of Lagrangian tendencies along backward trajectories initialized at (a, b) 250, (c, d) 500 and (e, f) 700 hPa in the last 24 hours before arrival in the cyclone area of (a, c, e) latitude and (b, d, f) longitude relative to the movement of the cyclone (i.e., with the 24 h longitude and latitude changes of the cyclone center subtracted). Contours show present-day Lagrangian tendencies, and the color shading indicates the response to future climate change (difference in the Lagrangian tendencies between future and present-day climate).

Minor comments
1. Line 103. Should 'proving' be 'providing'?

We will modify 'proving' to: providing

2. Line 148. If averaging over the entire cyclone area leads to cancellation between ascending and descending airstreams, why is this analysis presented? They also have a very large spread (line 176) meaning that interpretation of the averages is difficult.

We think that this analysis, although providing a relatively rough picture due to the averaging, is still useful for introducing the framework and giving first indications, for instance, of the relevant time scales. Despite the spread, we have shown differences between the lower and upper levels trajectories and estimated the time of most significant changes in several parameters to be 24 h before the initialization time. Furthermore, the basic effect of the warming climate becomes clear, that is an increase in potential temperature and specific humidity.

3. Figure 2. Is the shading around the present-day average the grey or red shading?

The gray shading corresponds to the present-day average. We will modify in the caption as follows:
Temporal evolution of (a,b) pressure, (c,d) latitude, (e,f) longitude, (g,h) specific humidity, (i,j) potential temperature and (k,l) PV averaged over all trajectories initialized within a 10° radius around the cyclone center of all selected cyclones and at (left column) 700 hPa and (right column) 250 hPa. The average for present-day climate is shown as blue, dashed line, the average over the future time slice as red line. The 5. and 95. percentiles are shown in gray shading for present-day and red shading for future climate.

4. Line 187: In the 24 h before what?

We will change this sentence to:
Trajectories reaching the cyclones at 700 hPa experience a clear PV increase in the 24 h before the maximum intensity.

5. Line 220. I suggest that the trajectories from the north have smaller absolute meridional displacement because the cyclone's themselves are typically travelling northwards enhancing to the airstream trajectory component in that direction (see major comments).

Yes, this is correct, as shown in the new Fig. R1. We will add a comment to the revised manuscript.

6. Line 223. I suggest that the relatively small region of westward displacement would be more significant if cyclone-relative longitudinal tendencies were plotted. This would give a better indication of cyclonic wrap-up of the air around the cyclone centre.

Yes, see again Fig. R1. We will add a comment to the revised manuscript.

7. Figure 5 and others. I think the description of blue and red lines should also be in the figure caption.

We will modify the caption for figures 4-6 by adding:  The average for present-day climate is shown as blue, dashed line, the average over the future time slice as red line.

8. Line 281.'Righ' should be 'right'.

We will modify 'Righ' to: right

9. Line 282. 'th' should be 'the'.

We will modify 'th' to: the

**References**

Binder, H., Joos, H., Sprenger, M., and Wernli, H. (2023). Warm conveyor belts in present-day and future climate simulations – Part 2: Role of potential vorticity production for cyclone intensification, Weather and Climate Dynamics, 4, 19–37, https://doi.org/10.5194/wcd-4-19-2023.

Catto, J. L., Shaffrey, L. C., & Hodges, K. I. (2010). Can climate models capture the structure of extratropical cyclones?. Journal of Climate, 23(7), 1621-1635.

Dacre, H. F., Hawcroft, M. K., Stringer, M. A., & Hodges, K. I. (2012). An extratropical cyclone atlas: A tool for illustrating cyclone structure and evolution characteristics, B. Am. Meteorol. Soc., 93, 1497–1502.

Fluck, E., & Raveh-Rubin, S. (2023). Dry air intrusions link Rossby wave breaking to large-scale dust storms in Northwest Africa: Four extreme cases. Atmospheric Research, 286, 106663.

Joos, H., Sprenger, M., Binder, H., Beyerle, U., and Wernli, H. (2023). Warm conveyor belts in present-day and future climate simulations – Part 1: Climatology and impacts, Weather and Climate Dynamics, 4, 133–155, https://doi.org/10.5194/wcd-4-133-2023.

Madonna, E., Wernli, H., Joos, H., & Martius, O. (2014). Warm conveyor belts in the ERA-Interim dataset (1979–2010). Part I: Climatology and potential vorticity evolution. Journal of climate, 27(1), 3-26.

---

## Author Comment (AC2)

**Answer to Reviewer 2**

This paper is the second part of a study looking at the future changes of extratropical cyclones in the CESM model. In this part the authors use Lagrangian trajectory analysis to investigate the pathways of the air parcels and changes in their characteristics on their way to different horizontal and vertical locations in composites of extratropical cyclones.

The results mostly corroborate the findings of the earlier paper, and of other studies, in finding that increased moisture in a warmer climate leads to increased diabatic heating and therefore larger PV production in mid levels. The upper level features are more complex, especially due to the level of focus often being above the tropopause.

I especially like the composite figures showing the tendencies over the past 24 hours, which gives a good understanding of the features of the cyclones.

I have a few comments that I hope might improve some aspects of the manuscript.

We are grateful to the Reviewer for their constructive comments, which will improve the quality of our manuscript. We are pleased to share our answers in this document. The figure and line numbers correspond to the original manuscript. The reviewer comments are in black and our responses are highlighted in blue.

General comments

I wonder at the choice of 250hPa as a level to focus on. This was considered in paper 1 also, but in that paper it is shown that the tropopause average pressure is close to 300 or even 350hPa. If the focus of the study is on the dynamics of the cyclones themselves, then would it not be better to look at a level within the troposphere, where the WCB outflow is having a more direct impact? This also makes the average of the trajectories over the cyclone area at the this level difficult to interpret, and possibly not very useful.

We have reproduced the figure for the 300 hPa level (see Fig. R2 below), resulting in a similar pattern with slight differences. For instance, in the present-day climate, there is a stronger change in pressure, indicating a stronger ascent of the WCB trajectories. The PV tendency composite shows a more evident PV decrease downstream.
However, since our main goal is to provide further insights into the processes shaping the PV anomalies shown in part I, which have been presented on the 250 hPa level we would like to keep the 250 hPa level also in this second part of our study. Furthermore, this is also consistent with other studies (e.g., Priestley & Catto, 2022).

[Figure]

Figure R2. Similar to Figures 3 and 7 but at 300 hPa.

Figs 4, 5, 6: In the section describing these figures, there is a lot of jumping around between these and the horizontal composites. This is because the 700hPa horizontal composites are discussed along with 4, 5, 6, then the 250hPa composites. It might be easier to read and follow if the trajectories from 700hPa are all combined into a single figure that can be discussed with the 700hPa horizontal composites. Then the same for the 250hPa trajectories. The way it is currently presented gives a slightly misleading impression that trajectories at different levels but the same location are more strongly related than they really are.

This is a good suggestion. We will modify the figures as suggested.

More specific comments

Line 12-14: This sentence is hard to read - consider rewording.

We will reword the sentence as follows:
In contrast, projected upper-level PV changes are due to a combination of several processes. These processes include cloud-diabatic PV changes, anomalous PV advection, and likely also radiative PV generation in the lower stratosphere above the cyclone center. For instance, enhanced poleward advection is the primary reason for a projected decrease in upper-level PV anomalies south of the cyclone center.

Line 31: Remove the additional comma.

We will remove this additional comma.

Figure 1: Since you later discuss the two branches of the WCB, I suggest adding the cyclonic branch onto this schematic.

We will add the cyclonic WCB branch to the satellite figure.

In the Methods section I would like a bit more information. I understand this is part 2, but it would be good to have some extra information so the paper can stand alone. For example, Which months? What area? NH or North Atlantic? How many storms does this make? Which cyclone identification?

We have added new information in the Methods:

3 Methods
We study Lagrangian airstreams in the 1% strongest cyclones in the 10-member CESM-LE dataset for the extended winter season (from October to March). This cyclone dataset is described in detail in part 1. Based on the SLP contouring method (Wernli and Schwierz, 2006), we identify and track storms over the North Atlantic region (longitude: -100° to 40° and latitude: 30° to 90°). The cyclone intensity and, thus, the extreme cyclone selection (1% strongest cyclones) are obtained by computing the relative vorticity at 850 hPa at the cyclone center. The number of extreme cyclones is 358 in the present-day and 308 in the future climate. In present-day climate, the cyclones typically travel towards the northeast, with the peak cyclone frequency south of Greenland. At the end of the century, the storm track is projected to shift eastward, implying a higher impact in the north of the United Kingdom and the west coast of Scandinavia.

Line 147-148: This argument only really works for the 700hPa trajectories, since at the higher level the trajectories are not so likely to be coming from above.

We see some trajectories coming from above also at 250 hPa, but of course not as many as at lower levels. We will add a note that this argument refers mainly to the 700 hPa level.

Line 167: Typo in the units.

We will modify g Kg$^{-1}$ to: g kg$^{-1}$

Figures 4, 5, 6: It may be nice to include the 5-95th percentile range on these figures too.

We will include the 5-95th percentile range to the Figures 4, 5, and 6.

Line 246: I find this more northward motion very interesting. Is this associated with a more poleward propagation of the cyclones? Or can you not infer that from this information? Is it possible to explain this feature in more detail?

The enhanced northward motion is also evident if the cyclones' propagation is removed (see the new figure R1e), albeit with a slightly lower magnitude. It is thus related to both the enhanced poleward propagation, which is consistent with previous studies (e.g., Tamarin & Kaspi, 2016) and a stronger northward flow of the air parcels relative to the cyclone center. A note on this will be added to the manuscript.

Line 249: Similarly to the previous comment, I find it interesting that there is a weakening in the westerly flow. It would be good to link this to projections of a weakened Jetstream either here or in the conclusions/discussion.

The weakening is restricted to lower levels. The tendency at upper levels is to strengthen the westerlies in the region of the jet streak (Fig. 7b).

Line 381: Typo in "cyclones".

We will change cyclonce to: cyclone

Figure S1: More information is needed in the caption - what level is this showing?

We will add more information, see below:

Figure S1. PV climatology at 300 hPa for the extended winter (October to March) in the North Atlantic region in present-day climate, future climate and their difference (response to climate warming).

**References**

Priestley, M. D. K. and Catto, J. L. (2022). Future changes in the extratropical storm tracks and cyclone intensity, wind speed, and structure,Weather and Climate Dynamics, 3, 337–360, https://doi.org/10.5194/wcd-3-337-2022.

Tamarin, T. and Kaspi, Y. (2016). The poleward motion of extratropical cyclones from a potential vorticity tendency analysis, Journal of the Atmospheric Sciences, 73, 1687–1707, https://doi.org/10.1175/JAS-D-15-0168.1.

---

## Author Comment (AC3)

**Answer to Reviewer 3**

This manuscript uses back trajectories to better understand how the potential vorticity structure and thus the dynamics of extra-tropical cyclones will change in a warmer climate. The main conclusions are that in a warmer and moister climate, enhanced ascent and latent heating in warm conveyor belts leads to a stronger low-level PV anomaly. In contrast, upper-level PV anomalies response in a much more complicated manner which the authors show is due to changes in advection and hypotheses that changes to radiative cooling near the tropopause are also important. Overall, the manuscript is well written, easy to follow, and the conclusions are well supported by the presented evidence. I have two concerns with this manuscript which I describe below, and numerous rather minor comments also listed below.

Thank you for reading our manuscript and for your constructive comments, which will help us to better communicate our results. In the following, we reply to your points. The figure and line numbers correspond to the original manuscript. The reviewer comments are in black and our responses are highlighted in blue.

Major comments:
1. Almost all of the results are presented as averages over all extreme cyclones. Cyclones are highly variable in their structure and dynamics. The impact of this variability is not taken into consideration in this study. Specifically:

a) Line 137-138 "we evaluate various parameters averaged over all trajectories initialised in the cyclone area, in a radius of 10 degrees around the SLP minimum" – this is a huge area and includes air masses with very different properties e.g., the cold sector, the warm convector belt. This huge variability is seen in Figure 2. Does this make scientific sense to average so many different trajectories together? However, in the other extreme, the authors then proceed to show trajectories from just one grid point (Figures 4, 5 and 6) which potentially are not representative. I strongly encourage the authors to re-consider their approach as I expect that much clearer and informative results may be obtained if trajectories only from certain areas of the cyclone were averaged together. Another recommendation, if the approach in Figures 4 – 6 is kept, is to include a measure of uncertainty on these figures, similar to what is done in Figure 2.

It is correct that cyclones can be highly variable and event-to-event differences may play an important role, e.g., for assessing the impacts of an individual storm. Nevertheless, in part I of this study we have shown that, also when averaging over many extreme cyclones, there is a *systematic* change in some of their properties in a simulated future climate, which warrants further investigation. We thus consider it as a valid and important approach to focus on the explanation of such mean changes also in this second part of the study.
We agree that the trajectory plots shown in Fig. 2 and those in Figs. 4-6 represent two "extremes" of the spectrum of possible analyses, but we would argue that the Lagrangian composites shown in Figs. 3 and 7 (and the cross sections in Figs. 8-10) fill the gap between these extremes. We think that Fig. 2, although providing a relatively rough picture due to the averaging, is still useful for introducing the framework and giving first indications, for instance, of the relevant time scales. Despite the spread, we have shown differences between the lower and upper levels trajectories and estimated the time of most significant changes in several parameters to be 24 h before the initialization time. Furthermore, the basic effect of the

warming climate becomes clear, that is an increase in potential temperature and specific humidity. The Lagrangian composites then directly provide what the reviewer is asking for: they show the spatial variability of the Lagrangian changes in the cyclone region, without the need to predefine specific regions for spatial averaging. The trajectories in Figs. 4-6 serve as illustrative examples of these changes, and their spatial representativeness can again be determined from the Lagrangian composites in Fig. 3.

We will include the 5-95[th] percentiles as shading also in the figures showing the trajectories from individual locations. Also for those, there is substantial variability due to the fact that more than 300 cyclones are considered. This variability would further increase if we'd average over a region.

   b) How do the magnitudes of the changes detected relate to the amount of variability in the control simulation? Or stated another way, are these results statistically significant? Figures 3 and 7 should include information showing where the changes are significant.

We will include stippling in the Lagrangian composite plots indicating where more than 80% of the ensemble members agree on the sign of the projected change.

2. Section 3. Some additional details of the simulations should be added here as it is not reasonable to expect a reader to read part 1. Even some basic information such as what time periods the simulations cover (this is in the abstract but could be repeated here), what resolution the simulations are performed at (the coarse resolution is noted as a limitation of this study in the conclusions, but a reader is not told what it is) would be appreciated. I also suggest that a few more details are given about the strongest 1% of cyclones – how many cyclones are there in absolute numbers in both the historical and future climate simulations? Do they all occur in a certain part of the north Atlantic or do they cover a huge geographic area? What metric is used to measure intensity?

We will add more information in the data and methods sections as follows (new information is highlighted in yellow):

2 Data
We have selected ten members from the CESM-LE-ETH ensemble, which were restarted from CESM-LE simulations (Kay et al., 2015) proving 6 hourly output fields on model levels that are required for our trajectory calculations (see section 3). The periods 1990-2000 (present-day climate) and 2091-2100 (future climate, under the RCP8.5 scenario) are analyzed. This fully coupled model has a horizontal resolution close to 1 degree (~0.94º in latitude and 1.25º in longitude). More details are provided in Sect. 2 of part 1.

3 Methods
We study Lagrangian airstreams in the 1% strongest cyclones in the 10-member CESM-LE dataset for the extended winter season (from October to March). This cyclone dataset is described in detail in part 1. Based on the SLP contouring method (Wernli and Schwierz, 2006), we identify and track storms over the North Atlantic region (longitude: -100° to 40° and latitude: 30° to 90°).  The cyclone intensity and, thus, the extreme cyclone selection (1%

strongest cyclones) are obtained by computing the relative vorticity at 850 hPa at the cyclone center. The number of extreme cyclones is 358 in the present-day and 308 in the future climate. In present-day climate, the cyclones typically travel towards the northeast, with the peak cyclone frequency south of Greenland. At the end of the century, the storm track is projected to shift eastward, implying a higher impact in the north of the United Kingdom and the west coast of Scandinavia.

Minor comments:

1. Line 52. Units Wm-2 is missing the negative sign.

We will modify $Wm^2$ to: $Wm^{-2}$

2. Line 58. "This PV ascent and descent"... This is rather strange, suggest revising it.

We will change this sentence to:
These PV changes are due to …

3. Line 69 – 70. This second branch of the cold conveyor belt is never mentioned again in the results section / the analysis so does it really exist on average or is this a rare feature?

The cold conveyor belt is usually difficult to distinguish, especially at the cyclone mature stage, because it can merge with the WCB cyclonic branch. It is not evident in our Lagrangian composites and thus not further discussed in the manuscript. We will add a corresponding note to the introduction.

4. Line 163, these values of specific humidity seem to be very small, however, it may be due to the large area that they are averaged over. Is this a valid hypothesis?

The 95th percentile is slightly above 4 g/kg, so spatial variability does play some role. Other reasons are the decrease of specific humidity with height (recall that we are looking at the 700 hPa level) and the fact that the cyclones typically reach their maximum intensity relatively far north.

5. Lines 170 – 200. This section discusses many of the processes we would expect in the warm conveyor belt, yet the results being discussed include all of the cyclone areas. This section should at least reminder a reader that the average trajectories also include those arriving in the cold sector.

This could be associated with the changes in the WCB trajectories being stronger than the other airstreams, having a more predominant signal. We will add the following sentence at line 183 to clarify that we are considering the trajectories arriving in the whole cyclone area: Recall that we have averaged the trajectories arriving in the cold and warm sectors.

6. Line 210 – could the location of these points be added to a composite map?

Yes, we will add markers to the composite plots.

7. Figures 3 and 7. The units on the colour bar on panel (c) are missing. It might also be a good idea to state in the caption here how many cyclones these composites were created from.

The units are provided in our PDF version of the preprint.

We will modify the captions and include the number of storms:

Figure 3. Composites of Lagrangian tendencies along backward trajectories initialized at 700 hPa in the last 24 hours before arrival in the cyclone area of (a) latitude, (b) longitude, (c) pressure, (d) potential temperature and (e) PV. Contours show present-day Lagrangian tendencies and the color shading indicates the response to future climate change (difference in the Lagrangian tendencies between future and present-day climate). A total of 358 and 308 storms are considered in the present-day and future climate, respectively.

8. Line 282 – typo "th" → the

We will modify th to: the

9. Figure 7e. There is a small area of negative PV tendency in the control simulation. I don't think this is discussed in the text. Is this related to negative PV tendencies about the localised heating maximum in the warm conveyor belt?

Yes, this is likely related to the PV destruction that the trajectories experience when reaching the upper levels. We will add a note on the negative values around line 300, where we discuss the corresponding process.

10. Line 310 – 327. There are many references to figures / results in part 1. This makes it quite difficult for a reader to follow without going to find the figures in part 1. Could this be revised so a readers' understanding does not require part 1?

Since the main goal here is to explain the PV anomalies identified in part I, we think that it is necessary and useful to refer to the respective figures from part 1.

11. Line 405-406. Can the references to the figures be added here? e.g., figure 2 for the Lagrangian composite at 700hPa.

Yes, we will modify the sentence to:
The Lagrangian tendency composite at 700 hPa (Fig.3), the north-south vertical cross-section through the cyclone center (Fig. 8c,d) and the time series in Fig. 4 (left column) show a descending airstream south of the center with characteristics of a dry intrusion (Raveh- Rubin, 2017). To study the spatial pattern of this DI in more detail, we analyze the west-east vertical cross-section (4) 4o south of the cyclone center, as shown in Fig. 10.

12. Line 424. Could add here what intense really means e.g. top 1% which is X numbers of

cyclones.

Yes, we will add more details, see below:

In this study, we have used a Lagrangian perspective based on air parcel trajectory calculations to investigate projected future changes of air streams in intense North Atlantic extratropical winter cyclones as well as their role for PV anomaly changes. The 1% strongest cyclones are considered, amounting to 358 cyclones in the present-day and 308 cyclones in the future climate.